# Breeding and Maintenance of Immunodeficient Mouse Lines under SPF Conditions—A Call for Individualized Severity Analyses and Approval Procedures

**DOI:** 10.3390/ani11061789

**Published:** 2021-06-15

**Authors:** Thomas Kammertoens, Sarah Jeuthe, Heike Baranzke, Antonina Klippert, Christa Thöne-Reineke

**Affiliations:** 1Institute of Immunology, Charité Universitätsmedizin, Campus Berlin Buch, Robert-Rössle-Str. 10, 13125 Berlin, Germany; thomas.kammertoens@charite.de; 2Max Delbrück Centrum Berlin, Campus Berlin Buch, Robert-Rössle-Str. 10, 13125 Berlin, Germany; 3Faculty of Humanities and Cultural Studies—Catholic Theology, Bergische Universität Wuppertal, Gaußstraße 20, 42119 Wuppertal, Germany; baranzke@uni-wuppertal.de; 4Nuvisan Innovation Campus Berlin GmbH, Muellerstrasse 178, 13353 Berlin, Germany; antonina.klippert@nuvisan.com; 5Animal Behavior and Laboratory Animal Science, Institute for Animal Welfare, Freie Universität Berlin, Königsweg 67, 14163 Berlin, Germany; Christa.Thoene-Reineke@fu-berlin.de

**Keywords:** immunodeficiency, severity assessment, animal experiment, 3R

## Abstract

**Simple Summary:**

In the EU, the breeding of genetically modified laboratory animals is defined as an animal experiment, if the animals may experience pain, suffering or harm due to the genetic modification. In most cases, modifications of the immune system do not lead to pain, suffering, or harm given the animals are kept under specific pathogen-free housing conditions. Nevertheless, the EU working group on genetically altered animals defined SPF housing conditions as “Refinement”. According to this working group, applied refinement strategies for breeding genetically modified animals do not exclude them from project licensing. Thus, theoretical suffering, pain or harm is assumed without applying methods of actual burden assessment. Furthermore, the definition of SPF as a refinement strategy rather than a standard husbandry procedure leads to incomprehension among the scientific community in the EU. Here, we discuss the ethical basis of animal experiments as well as the current legal situation for immunodeficient animals. Furthermore, we discuss potential pain, suffering, or harm of immunodeficient animals from an immunological perspective. Finally, we briefly outline an animal welfare-oriented approach to severity assessment of immunodeficient mice.

**Abstract:**

In the EU, the breeding of genetically modified laboratory animals is, by definition, an animal experiment if the offspring may experience pain, suffering, or harm. In order to determine the actual burden of genetically modified mice, established methods are available. However, the breeding of immunodeficient mice is considered an experiment requiring a permit, even if no pain, suffering or harm is observed under scientifically required defined hygienic housing conditions, as determined by established methods of severity assessment. This seems contradictory and leads to uncertainty among scientists. With this commentary, we would like to shed light on this topic from different perspectives and propose a solution in terms of individualized severity assessment and approval procedures.

## 1. Introduction

In the EU, breeding of genetically modified laboratory animals is, by definition, an animal experiment if the animals being born may experience pain, suffering or harm. To assess the actual pain, suffering, or harm in genetically modified mice, established methods are available and frequently used. The result of this severity assessment leads to the determination if a genetically modified strain is to be classified as burdened and its breeding thus continues to be an animal experiment requiring a permit. However, the situation for immunodeficient animals is different. The breeding of immunodeficient mice is considered an experiment per se requiring a permit, even if no pain, suffering or harm is observed in the severity assessment under the scientifically required defined hygienic housing conditions. This corresponds to the assumption of classifying these housing conditions as refinement which, in turn, does not relieve from the approval requirement. This deviation from the standard severity assessment procedure for immunodeficient animals is, however, not directly comprehensible from a scientific point of view. In this interdisciplinary commentary, this topic will be examined from the different perspectives of an ethicist, an immunologist, a laboratory animal scientist, and a legal point of view. While this commentary is written primarily from a German and European perspective, the underlying issues of how best to employ laboratory animal science to contribute to the protection of animals and to an appropriate legal framework are of general interest.

## 2. Animal Experiments a Question of Conscience—The Ethical Perspective (Heike Baranzke)

Animal experiments are highly controversial in terms of animal ethics—not only in terms of assessing what is legally permitted (legal), but what is morally justifiable (legitimate) as well. However, according to which criteria should this be decided? A good clue is provided by the purpose statement of the German Animal Welfare Act (TierSchG) as the moral goal to be realized by the following individual legal norms. The statement indicates that the Animal Welfare Act is not intended to encourage people to find loopholes in what is forbidden, but to read the legal norms as a guideline in fulfilling the purpose of the law, which is “to protect the life and well-being of animals out of man’s responsibility for them as fellow creatures. No one may inflict pain, suffering or harm on an animal without reasonable cause” (§1 TierSchG). The obligation to protect the life and well-being of the animal and the provision not to inflict pain, suffering, or harm on the animal without reasonable cause outline the never-ending moral task of honest striving for the avoidance of any unnecessary pain, suffering, or harm imposed on animals used for human purposes—here focusing on higher animals, especially vertebrates and cephalopods. The recognition of the animals’ capacity to experience pain and suffering as psychophysical living beings is crucial and was not taken for granted for a long time. In the 17th century, the philosopher and physicist René Descartes had declared animals to be unfeeling mechanical automata. This position was used to legitimize scientific vivisection and, at the beginning of the 20th century, was further supported by an ideologically exaggerated behaviorism and a reductionist reflex theory that denied any psychological association of the animal. The animal’s capacity for pain and suffering was rejected as unscientific anthropomorphism. The recognition of animals as sentient beings capable of suffering is the hallmark and criterion of so-called pathocentric animal protection and a hard-won milestone of the animal protection movement against mechanistic views on animals.

However, we cannot speak of the “animal” without at the same time speaking of the “human being”. As human beings, we understand ourselves as moral beings free to determine our will and our actions in order to lead a good life, morally and physically. In the context of biomedical ethical debates, we have come to understand ever more deeply how much our own psychophysical state of mind is part of our idea of a good life, which involves the freedom from pain and suffering. While the capacity for moral responsibility is a human-specific characteristic, with which we differ from the animal (singular), we are similar to the other animals (plural) with regard to the capacity of pain and suffering. In moral terms, this, however, implies that it would be contradictory to want to strive for freedom from pain and suffering as a moral good for ourselves while ignoring the same ambition in other living beings. Facing this basic moral conflict again and again and determining the “reasonable cause” to inflict “pain, suffering or harm” on an animal in the sense of the Animal Protection Act § 1 is a question of conscience that cannot be avoided. To summarize: Cultivating one’s empathy with the animal as a sentient being and striving for sincerity regarding the assessment of the burden caused by the physical manipulation and regarding its justification by the purpose of research strengthens credibility in the debate concerning objections to animal experimentation and may be a stress test going beyond the minimum legal requirements. This central idea is taken into account, for example, in the context of the ethical justifiability within the approval procedure for animal experiments.

## 3. No Compelling Evidence to Support a Link between Immunodeficiency and Burden on Laboratory Animals in the Absence of Specific Pathogens—The Immunological Perspective (Thomas Kammertoens)

The immune system is a complex network consisting of a wide variety of immune cells that can be found in all organs of the body or recruited via blood and lymph. Immune cells communicate with all cells of the body via mediators or surface molecules. In this manner, they perform regulatory and surveillance functions that ensure an appropriate response to microbial challenges and maintain homeostasis of body functions. Our understanding of the immune system is based, in large parts, on animal experiments. Phagocytes have been discovered in starfish [1], T cells in mice [2], B cells in birds [3], and the toxin-neutralizing serum activity that Paul Ehrlich attributed to “Antikörper” (antibodies) was discovered in rabbits [4]. Similarly, discoveries of clinical importance, such as the development of monoclonal antibodies, for example, are based on mouse experiments [5]. Although new technologies such as “in silico” models, 3D organoids, as well as advances in human immunology and novel “omics” technologies will allow to reduce or replace animal experiments, mouse models for studying the complex interaction of the immune system with the organism over time continue to be an irreplaceable asset in immunological research. Especially in preclinical research, such as the development of the SARS-CoV-2 vaccine, animal experiments are essential [6] and often required by law before the experiment can be conducted in the human system (i.e., the clinical trial can be initiated). Immunological research often uses genetically modified mice, which are generally kept under specified pathogen-free conditions (SPF, i.e., free of predefined specific pathogens). Of note, modern husbandry systems (individually ventilated cages, transfer of the animals under sterile workbenches, etc.) allow for breakdowns in the hygiene regimes to be ruled out with great certainty.

In 2010, the Directive 2010/63/ EU of the European parliament and of the council on the protection of animals used for scientific purposes (Directive 2010/63/EU) was released. It stipulates that the breeding of transgenic animals be subject to authorization if the genetic modifications may cause pain, suffering, or harm. In 2013, a working group was tasked with the interpretation of the directive within which two basic assumptions are untenable from an immunological point of view: (1) the working group defined a specific pathogen-free (SPF) animal husbandry as refinement. This is not tenable from a scientific point of view. The exclusion of all known pathogens (according to FELASA) is a condition sine qua non for all (also immunocompetent) laboratory animals and thus, is simply considered good scientific practice. (2) The assumption that it is possible to determine the susceptibility to infection for genetic alterations affecting the immune system in advance does not apply to the vast majority of genetically altered animals used in immunological research any more than it does to immunocompetent animals. Nevertheless, based on the definition of the working group, breeding and keeping of animals with a so-called “modulated immune system” became subject to licensing due to a risk of infection perceived by the working group. Accordingly, at present for example in Germany, the breeding of laboratory animals falls under the legal definition of an animal experiment (according to § 7 para. 2 p. 1 no. 2 Animal Welfare Act) if the animals could theoretically experience pain, suffering, or harm. Consequently, the breeding of mice with genetic alterations that may affect the immune system must be applied for prophylactically as an animal experiment [7]. However, in our own experience of many years and also in the experience of many colleagues from the Animals in Research Committee of the German Society for Immunology (DGFI), no pain, suffering, or harm is typically observable in immunodeficient animals under the scientifically mandatory SPF conditions. With very few exceptions, for example the inoculation of immunocompetent mice with defined pathogens—so-called “wildlings”—to create mice with a certain type of microbiom [8], SPF maintenance of all mice, regardless of the genotype, is required to ensure reproducible results. 

The contrast between the observed lack of a burden of most genetically modified animals used in immunological research under SPF housing conditions and the licensing requirement for all these lines often leads to incomprehension among immunologists, since—from their point of view—no added value for animal welfare, but only an increased bureaucratic effort arises. For immunologists, there is no comprehensible reason to proceed differently in immunological research than with other genetically modified mouse lines, namely to first consider if there are data available that clearly indicate a burden to be expected and, if not so, to observe whether a burden occurs under the respective breeding and husbandry conditions and to initiate the licensing procedure if some sort of burden is associated with the genotype.

Evidence accumulates that complex interactions between the immune system and the nervous system exist also in the steady state, in the absence of infections (for a comprehensive review see [9]). However, it is not yet clear if learning disabilities and cognitive deficits—associated, for example, with altered cytokine or immune cell regulation—affect the animal in a positive or negative fashion. Since no in-depth studies on the burden of immunodeficient mice in the absence of pathogens have been conducted, a recent study analyzed this issue. In genetically modified mouse strains with severe immunodeficiency, i.e., a knockout for Recombination Activating Gene 2 or lacking expression of interferon gamma receptor, it was shown that a burden does not necessarily occur under SPF conditions [10]. Nevertheless, as explained above, the legal situation requires the breeding of both above-mentioned mouse strains to be prospectively classified as subject to approval, since—in the view of the working group of the Directive 2010/63/EU—pain, suffering, or damage in the offspring cannot be reliably ruled out in advance. The EU Commission working group’s argument is that SPF husbandry represents a measure of refinement (i.e., a technical measure, in the sense of the 3R concept, which is undertaken exclusively to reduce the burden) and that a burden cannot be completely ruled out. In this context, it is assumed that in case of potential hygiene breakdowns, the welfare of immunodeficient mice will be more severely affected than that of wild-type animals. This argument is, however, not conclusive from an immunological point of view. Scientifically, also most non-genetically modified mice with a natural immune system need to be kept under SPF conditions in immunological experiments in order to prevent bias. The use of hygienically defined animals is therefore simply regarded “good scientific practice” for immunologists, as is, for example, the use of hygienically defined, mycoplasma-free cell cultures. 

The assumption that immunodeficient animals are burdened per se and that the risk of infection is predictable is problematic for several reasons: Due to the complexity of the interaction of the immune system with the organism and the respective microbes, it is implausible to predict the role of individual genes in the outcome of an infection without empiric evidence. For example, fully immunocompetent BALB/c mice die from infection with certain strains of Leishmania, whereas immunocompetent C57BL/6 mice and putatively immunodeficient BALB/c interleukin-4 receptor deficient mice can cope with infection [11]. As can be observed for the current SARS-CoV-2 pandemic, there are also large differences in humans with respect to immune responses against a given pathogen. A wide variety of clinical courses have been described, ranging from asymptomatic to mild or even severe. It should be noted that, in the vast majority of cases, these are immunocompetent individuals, but they differ in terms of allelic variants of the gene-pool in the population and in terms of environment. 

Even humans with mutations in central immunological control circuits, such as the type 1 interferon signaling pathways, can live in a “normal” environment (that contains pathogens) for decades and only when exposed to a specific pathogen (such as certain viruses) do they experience severity [12,13]. Finally, as also illustrated in the example of the current SARS-CoV-2 pandemic, pathogens change randomly during evolution—therefore, absolute predictions of immunocompetence and deficiency are problematic.

All these examples illustrate that the degree of immunocompetence to pathogens usually cannot be predicted for both the genetically non-engineered organism and the organism with a genetically modified immune system. 

Finally, the terms employed by the document of the working group interpreting the Directive 2010/63/EU and the ensuing discussion describing the animal models used in immunological research are too broad and scientifically undefined; for example, “immunomodulated”, “immunocompromised,” or “immunodeficient” are mentioned. These broad umbrella terms, which together may include more or less all genetically modified mice and inbred mouse strains, do not allow for a consistent conclusion on the impact of the genetic change on the organism’s well-being or resistance to infection.

Taken together, from an immunological perspective, there is no compelling evidence to support a link between immunodeficiency and burden on laboratory animals in the absence of specific pathogens. Therefore, the current guidelines to the Directive 2010/63/ EU stated by the working group in 2013 are not based on empiric evidence and scientific reasoning. Also, the premise that defined hygienic housing conditions are not used as a scientific standard for reproducibility but are rather solely installed as refinement for immunodeficient mice is ill-conceived. From a 3R perspective, an individualized severity assessment allowing for a precise analysis under given environmental and housing conditions seems to better account for animal welfare than a generalized severity assumption lacking empiric data. If immunodeficient animals were treated as all other genetically modified animals with an individualized severity analysis, this would allow for a refinement solution based on the actual burden under the given housing conditions.

## 4. Legal Background on the National and the European Level—A Legal “Balancing act” (Antonina Klippert) 

When conducting animal experiments, there are several legal levels that have to be considered: the Directive 2010/63/EU on the protection of animals used for scientific purposes, the German Animal Welfare Act (TierSchG), the Animal Welfare Experimental Animals Ordinance (TierSchVersV), and further guidelines from the European Commission. Their relationship, position, and complex interplay must be considered when interpreting animal experimentation law—a legal balancing act that becomes particularly apparent when problems arise from animal experimentation law practice. 

The basis at European level is the Directive 2010/63/EU, which had to be transposed into national law by each member state. The instrument of the directive enabled the member states to embed the requirements in preexisting national law. However, the implementation must be in strict conformity with the Directive. If ambiguities exist in the interpretation of the law on national level, the requirements of the Directive must also be considered [14]. 

In general: Breeding of genetically modified lines requires project authorization if the animals may experience pain, suffering, or harm due to their genetic modification. With regard to the legal classification of the breeding of genetically immune-modified animals, there is still uncertainty among the scientific community in Germany as to whether their breeding requires project authorization. The National Committee for the Protection of Animals Used for Scientific Purposes for the Federal Republic of Germany has therefore established a legal classification [7].

An animal experiment, as defined by § 7 para. 2 TierSchG, takes place if (1) procedures are carried out (2) for experimental purposes that (3) may result in animals possibly experiencing pain, suffering, or harm. The term “animals” includes both animals that directly undergo the procedures or animals that are not yet born, but could suffer after birth due to the genetic modification. Only in cases in which all these criteria are fulfilled, the breeding of genetically modified animals is considered an animal experiment.

In this context and in the sense of Art. 3 No. 1 of the Directive 2010/63/EU, a procedure is carried out, since non-invasive procedures such as mating of animals are also included in the definition. According to Art. 3 No. 1 sentence 2 of the Directive 2010/63/EU, the purpose of the experiment also includes “other scientific purposes”, as is the case for the breeding of immune-modified animals for use in future experiments. The most controversial issue is whether the immune-modified animals may experience pain, suffering, or harm. It should be noted that the possibility of pain, suffering, or harm is already sufficient. If immune-modified animals are more susceptible to this possibility due to their genetic modification than, for example wild-type animals, the argument that animals without genetic modification are also at risk for infection during hygiene breaches can only be used to a limited extent. 

Therefore, pain, suffering, or harm cannot be excluded and all three prerequisites for the classification of an animal experiment according to § 7 para. 2 no. 2 of the German Animal Welfare Act are fulfilled. 

The arguments are supported by the EU working paper, which must be taken into account in the interpretation of the Animal Welfare Act whenever uncertainties arise. Here, it is specified beyond doubt that the breeding of animals with a modified immune system should be covered by the authorization requirement: “*Genetically altered lines which retain a risk of development of a harmful phenotype (e.g., (...) risk of infection due to compromised immune system) regardless of the refinement applied (e.g., barrier conditions (...), their breeding requires project authorization (...)”* [15]. In this respect, the interpretation of the German animal experimentation law is limited by the requirement of compliance with EU law. 

To summarize from a legal point of view, the breeding of immunodeficient genetically modified mouse lines in Germany constitutes an animal experiment. The reasons lie in the interpretation in conformity with the Directive 2010/63/EU and the interpretative notes of the European Commission, considering the definition of SPF husbandry conditions as a refinement. However, a more detailed legal definition by the EU of the term “refinement” and whether barrier housing in this particular case is included in that definition would be necessary. Thus, this topic could be reflected from a different perspective. 

Nevertheless, while taking into account the legal situation, it is important to incorporate scientific knowledge, which leads to a better understanding of all parties involved and ultimately serves harmonization. 

## 5. Severity Assessment and 3Rs (Sarah Jeuthe, Christa Thöne-Reineke)

In modern laboratory animal housing, the standard housing conditions for mice according to good laboratory animal science practice are barrier housing with individually ventilated cage systems and a hygiene concept in accordance with the recommendations of FELASA. Here, regular microbiological examinations of the laboratory animals and hygiene measures ensure the hygiene status of these animals during breeding, husbandry, and the experiment in order to protect laboratory animals from non-experimental diseases, to prevent a bias of the experimental results, and to contribute to animal welfare. Based on these premises, barrier housing is not considered refinement, but a “state of the art” housing condition. These husbandry conditions must also be demonstrated to the regulatory authorities as part of the approval procedure for animal experiments and help to ensure pain, suffering, or harm are kept to an absolute minimum. Considering that these housing conditions are mandatory, it is difficult to comprehend why immunodeficient animals per se fall under the permission requirement and barrier husbandry is classified as refinement [7]. In the context of the approval procedure, the pathogen exposure of the animals used in experiments must be limited to an indispensable level. 

In the context of breeding, husbandry, and the generation of new transgenic mouse lines, evaluation procedures have been established [7], which are based on many years of experience and on the basis of which it can be assessed whether a breeding is burdened. In accordance with the recommendation of the National Committee in Germany, for example, animals of a transgenic line are routinely evaluated at different time points in the course of their lives. Criteria of this evaluation include, for example, the weight development and other phenotypic characteristics, which are then compared to wild-type animals. Of course, the 3R principles established in 1959 [16] also apply to ensure that mouse breeding and experiments are avoided wherever possible (replacement), the number of animals used is kept to a minimum (reduction), and stress is restricted to the indispensable level (refinement). The latter also includes the animals being comfortable in an enriched environment, in which they can act out their species-typical behavior. 

Whilst genetically modified, immunodeficient mice were previously assumed a burdened line per se [15]; a study on severely immunodeficient mice showed that, depending on the respective mouse line and husbandry, no impairment of the well-being was detectable [10]. In view of these findings, the classification of burdened versus unburdened breeding is based on an individualized and differentiated strain assessment of the respective immunodeficient mouse line with regard to the respective animal facility. In contrast to the blanket burden classification, this differentiated approach would allow for the identification of breeding lines actually associated with an increased burden under given breeding conditions. Existing resources could then be better directed to these burdened breedings and used more specifically for animal welfare. Hence, immunodeficient lines, like all other genetically modified mouse lines, should be subjected to individual severity assessment and not generally classified as burdened, as SPF husbandry is considered a worldwide accepted standard in experimental animal husbandry and not a refinement.

## 6. Conclusions 

From a moral and legal point of view, animal welfare is of great importance to society. Scientists bear a special responsibility, as they must be able to assess the burden on laboratory animals. Together with animal welfare officers and licensing authorities, they are obliged to classify the burden in the course of the licensing procedure in advance, which is essential for balancing pain and suffering on the one hand and knowledge gain on the other. For the actual burden assessment, established methods are available to perform such a classification also for immunodeficient mouse lines. We postulate it is important for the EU Commission to consider barrier housing a husbandry prerequisite for mice, including immunodeficient animals, rather than a measure of “refinement”. Recent scientific publications show that there is no evidence of burden for certain immunodeficient strains under standard housing conditions. However, further studies are necessary to establish a more solid data base.

## Data Availability

Not applicable.

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
