# Peer review of "Breeding and Maintenance of Immunodeficient Mouse Lines under SPF Conditions—A Call for Individualized Severity Analyses and Approval Procedures"

_animals, 2021, doi:10.3390/ani11061789_

Round 1

Reviewer 1 Report

Because there are some description that are difficult to understand, English edition by the native speaker may be needed.

Author Response

Dear Reviewer,

We thank you and the other reviewers for your input into our commentary. This was very helpful to improve the commentary!

Based on your suggestions we have revised the manuscript that is entitled ” Breeding and maintenance of immunodeficient mouse lines under SPF conditions - A call for individualized severity analyses and approval procedures” by Kammertoens et al., to be considered for publication as a “commentary” in Animals.

You pointed out that the English needed to be improved. We have had a native speaker go over the text in order improve the language.

Best regards

Sarah Jeuthe

Reviewer 2 Report

This commentary addresses an important topic but, in my opinion, the style of writing needs to be simplified. There is a great deal of statement in each section but I feel that the development of cogent rationale in each section is lacking. I would like to see the development of simple take home messages in each section and in the conclusion.

I feel that this commentary needs to be put into context. It is mostly argued around the German situation, with reference to the European Union, but how applicable are the arguments internationally?

I feel that there needs to be definition of many terms, e.g. burdon, suffering, welfare, well-being and stress. Do the authors distinguish between welfare and well-being or do they consider them the same thing?

There is understandably a strong focus on the immune system but I feel that there is insufficient consideration of other aspects of animal functioning that contribute the welfare of immunodeficient mouse lines. In particular, mental state should be discussed in this commentary. I feel that this would strengthen the commentary.

Author Response

Dear Reviewer 2,

We thank you and the 2 other reviewers for your input into our commentary. This was very helpful t improve it!

Based on your suggestions we have revised the manuscript that is entitled ” Breeding and maintenance of immunodeficient mouse lines under SPF conditions - A call for individualized severity analyses and approval procedures” by Kammertoens et al., to be considered for publication as a “commentary” in Animals.

Two of  you pointed out that the English needed to be improved. We have had a native speaker go over the text in order improve the language.

We are thankful for the comments that helped us to correct and improve the manuscript.

Below please find a point by point reply to the critiques.

Best regards

Sarah Jeuthe

Point by point reply

This commentary addresses an important topic but, in my opinion, the style of writing needs to be simplified. There is a great deal of statement in each section but I feel that the development of cogent rationale in each section is lacking. I would like to see the development of simple take home messages in each section and in the conclusion.

We agree that the 4 very different perspectives are somewhat divergent. We thank the reviewer for the suggestion to briefly summarize each of the 4 sections and hope that our summaries convey the most important points as take home messages.

I feel that this commentary needs to be put into context. It is mostly argued around the German situation, with reference to the European Union, but how applicable are the arguments internationally?

We agree that the issue we discuss center around a scientific and legal controversy that causes Problems primarily in Europe. However as we have stated in the last sentence of the introduction of the revised version: “While this commentary is written primarily from a German and European perspective, the underlying issues of how best to use laboratory animal science to contribute to the protection of animals and an appropriate legal framework are of general interest”.

I feel that there needs to be definition of many terms, e.g. burdon, suffering, welfare, well-being and stress. Do the authors distinguish between welfare and well-being or do they consider them the same thing?

We agree that our terminology was not exact. As our issue primarily deal with the legal framework and animal welfare, we went through the text and now rather talk about pain, suffering and harm as well severity.

There is understandably a strong focus on the immune system but I feel that there is insufficient consideration of other aspects of animal functioning that contribute the welfare of immunodeficient mouse lines. In particular, mental state should be discussed in this commentary. I feel that this would strengthen the commentary.

We thank the reviewer for the comment and agree that the field of neuroimmunology and how the immune system affects the nervous system and behavoir is a fast devloping and exciting field of research, that will also affect laboratory animal science. We have added this thought in the third section of the commentary stating that: “Evidence accumulates that there are complex interactions between the immune system and the nervous system also in the steady state, in absence of infections (for a comprehensive review see [9]). However, it is not yet clear if learning disabilities and cognitive deficits associated for example with altered cytokine or immune cell regulation affect the animal in a positive or negative fashion.”.

For this complex field of neuroimmunology we have reffered the reader to the excellent review of our colleague Dr. Kipnis in Nature Reviews Immunology.

Reviewer 3 Report

Several scientists give their scientific point of view that the breeding of immunodeficient mice keeping in barrier system be subject to authorization should be In-depth and careful evaluation.

This is well prepared comment of manuscript, which discussed animal study using immunodeficient mice, law, animal welfare, and ethical issue.

Author Response

Dear Reviewer 3,

We thank you and also the two other reviewers for your input into our commentary. We are thankful for all comments!

Based on the reviewers suggestions we have revised the manuscript that is entitled ” Breeding and maintenance of immunodeficient mouse lines under SPF conditions - A call for individualized severity analyses and approval procedures” by Kammertoens et al., to be considered for publication as a “commentary” in Animals.

Two of you pointed out that the English needed to be improved. We have had a native speaker go over the text in order improve the language.

Additionally we have modified the manuscript according to the critiques from reviewer 2. 

Best regards

Sarah Jeuthe